# Effect of Dietary Supplementation of Black Cumin Seeds (*Nigella sativa*) on Performance, Carcass Traits, and Meat Quality of Japanese Quails (*Coturnix coturnix japonica*)

**DOI:** 10.3390/ani12101298

**Published:** 2022-05-18

**Authors:** Muhammad Umair Asghar, Sibel Canoğulları Doğan, Martyna Wilk, Mariusz Korczyński

**Affiliations:** 1Department of Animal Nutrition and Feed Sciences, Wroclaw University of Environmental and Life Sciences, 25 C.K. Norwida St., 51-630 Wrocław, Poland; martyna.wilk@upwr.edu.pl (M.W.); mariusz.korczynski@upwr.edu.pl (M.K.); 2Department of Animal Production and Technologies, Faculty of Agricultural Sciences and Technologies, Niğde Ömer Halisdemir University, 51240 Niğde, Turkey; scanogullari@ohu.edu.tr

**Keywords:** black cumin powder, meat quality, natural-antioxidant, performance efficiency, quail

## Abstract

**Simple Summary:**

Over the last decade, there has been a surge of interest in the use of natural herbs as antibiotic alternatives or natural feed additives in diets to boost animal productivity and optimize the potential production. One of the natural feed additives is black cumin powder (BCP), which is characterized by high antioxidant activity and high phenolic contents. The aim of this study was to evaluate the influence of varying levels of BCP added to the Japanese quail diet on the growth, slaughter carcass, sensory features, and some meat preservation properties. It is worth noting that BCP lowered the overall quantity of bacteria and increased the quality of meat preservation. According to our findings, the supplementation of BCP had a positive influence on the quail growth, lipid profile, antioxidant, immunity, meat storage quality, pH, and decrease in pathogenic bacteria content.

**Abstract:**

The current study was conducted to determine the effect of adding black cumin (*Nigella sativa*) powder (BCP) to the Japanese quail diet on the carcass characteristics and meat quality. In this research, 240 Japanese quail chicks (mean initial body weight 9.15 ± 0.12) were divided into four groups of four replications each. Treatments consisted of the addition of BCP at levels 1, 2, and 4% to the mixed feed and control group without the BCP additive. Compared to the other groups, the group with the 2% BCP diet had a higher live weight (LW), body weight gain (BWG), and a better feed conversion ratio (FCR, *p* < 0.05). BCP administration had no impact on the carcass characteristics, however, BCP had a significant effect on the thigh and breast meat. The animal study protocol was approved by the Niğde Governorship, Provincial Directorate of Agriculture and Forestry, Turkey (protocol code: E-15018773-050.01.04-75932 and date of approval: 26 April 2021) for studies involving animals. Lower thiobarbituric acid (TBA), pH, peroxide, and total psychrophilic bacteria levels were found in the BCP added groups compared to the control group (*p* < 0.05). When compared with the control, the sensory properties such as color, juiciness, softness, and flavor were significantly higher in the BCP treated groups, especially when fed the 2% BCP diet. It can be concluded that BCP as an additive to quail feeds had a significant effect on the performance of quails as well as on the shelf life of the meat. In order to avoid health and environmental concerns, it was concluded that BCP can be used as a natural additive to replace synthetic antimicrobials and antioxidants at the level of 1–2% in quail compound feeds.

## 1. Introduction

The poultry industry is one of the most lucrative production sectors around the globe that provides a high-biological value animal protein. The expected population growth is related to new challenges (e.g., increasing demand for food; 60% higher food production in 2050 compared to the current level) [1]. There is a present shortage of protein sources for animals, as a consequence, there will be decreased protein sources for humans. To overcome this protein gap, breeders will have to use synthetic chemicals in the poultry sector. Previously, antibiotics held a massive position in the industry, which were widely used in broilers as growth promoters. Nowadays, research efforts are being focused on reducing the use of antibiotics in livestock farming, especially in light of the bacterial resistance in human beings [2]. Various zoonotic infectious microorganisms such as *Salmonella*, *Escherichia coli*, and *Enterococci strains* can be abridged with the aid of synthetic chemicals in the diets of broilers. The dietary insertion of antibiotics in poultry feed as a growth promoter mostly leads to the occurrence of cross resistance amid pathogens and can also be a cause of residue tissues in the animal body [3]. As a result of this issue, the European Union (EU) forbid the routine practice of antibiotics as growth promoters in the diets of animals in January 2006 [4]. The antibiotic residues in meat and eggs [5] showed problems regarding human health [6], which is why antibiotics are cautiously used in the broiler sector to execute efficiency and raise the quality of carcasses with less ratio of fat and over amount of protein in the meat [7]. Concerning the food ingredients, nowadays, these are embattled for, not only quantity, but also in quality. As a result, animal breeders and consumers are interested in feed and food that contain enormous bioactive components along with better health advantages [8].

Today, scientists are searching for natural growth promoter ingredients such as essential oils, phytobiotics, and medicinal origin plants, which may be advantageous due to their antimicrobial resistance [9,10]. Such kinds of medicinal plants also show stimulating special effects on the digestive system of animals [11,12,13].

Because of the fat deposition or storage in the meat, consumers are prone to different kinds of diseases such as cardiovascular problems, cancer, and overweight [14]. In poultry meat, there exists an amplified amount of polyunsaturated fatty acids that are more prone to oxidative degradation. In meat cooking, the processing and cold storage units as well as the oxidation of lipids are the main obstacles. The oxidation of meat lipids leads to a decrease in the shelf life of meat, deteriorates the taste, the standards of the food, and also affects the organoleptic characteristics. To ensure the high-quality, the food products should be maintained by adding antioxidants (natural or synthetic) that do not permit the oxidation of lipids [15,16].

To obtain high-quality nutritious products, it is crucial to acquire them by using healthy ingredients and decreasing the use of unhealthful products [17]. Consumers are looking for products that have a low-fat and -sodium contents and products that are natural sources of antioxidants, antimicrobials, omega-3, and omega-6 fatty acids [14]. Aromatic plants play an extremely crucial role in animal feed due to the content of biological active compounds (e.g., flavonoids and essential oils—EOs) [18]. EOs enhance digestive enzymes, boost feed conversion ratio, provide antioxidant qualities, and underpin animal immunology. As reported by Akram et al. [19], the use of EOs in feed is a simple and effective way to improve livestock production. Biologically active compounds are present in different kinds of plants such as garlic, ginger, oregano, parsley, black cumin, curcumin, cinnamon, rosemary, etc.

Black cumin (*Nigella sativa*) contains an extraordinary amount of carbohydrates, protein, amino acids, lipids [20], calcium, potassium, phosphorus, and magnesium [21]. Moreover *Nigella sativa* contains saponins, alkaloids, volatile, oils, and a diversity of pharmacologically bioactive components such as thymoquinone, dithymoquinone, carvacrol, thymol, nigellicine-N-oxide, nigellidine, and α-hedrin [22]. Black cumin has the ability to reboot antimicrobial activities and antioxidative traits [23,24,25]. Moreover, black cumin has strong positive effects on the gastrointestinal tract and muscle relaxant effects [11]. Black cumin seeds can be used in feed as versatile growth promoters and might show extraordinary effects on the performance of broilers [26].

A total of 18 different kinds of compounds are present in black cumin including 99.15% of the total essential oil and cumin aldehyde (23%), gamma-terpinene (14.5%), acetic acid (10.9%), 1,3,8-p-acetic acid (10.9%), and 1,3,8-p-menthatriene (7.9%) [27]. The fundamental component of black cumin essential oil is carvone, which crafts about 66% of the oil and limonene, which accounts for 50% of the oil [28]. Further fundamental compounds of black cumin are sabinene, carvon, carveol, flavonoids, polysaccharides, coumarin, and cuminaldehyde, which have effects against fungi [29], bacteria [30], spasmodic [27], and inflammation as well as analgesic effects [31]. Amid the phytobiotic plants, black cumin is a well-notorious medicinal plant, which works against various bacterial diseases. Therefore, the dietary addition of black cumin powder in the diets of quails may show positive impacts on the quality and quantity of the meat. The main aim of the study was to explore the impact of black cumin powder on the performance, carcass characteristics, pH, color of the meat, lipid oxidation, and sensory chattels of quail meat.

## 2. Materials and Methods

The present research was conducted at Niğde Ömer Halisdemir University’s Ayhan Şahenk Agricultural Research Application and Research Center, Turkey. The laboratory analysis was performed at the Niğde Ömer Halisdemir University (NÖHU), Faculty of Agricultural Sciences and Technologies, Department of Animal Production and Technologies.

### 2.1. Animals

The hatchery in the quail unit of NÖHU, Ayhan Şahenk Agricultural Research Application and Research Center, provided 240 Japanese quail birds (mean initial body weight 9.15 ± 0.12). According to the fully randomized design, the birds were allocated into four treatment groups with four replicates, 15 birds each. Mixed-sex groups were put into quail cage compartments. The average weight of the birds at the beginning was kept constant.

### 2.2. Feed Material

All birds were fed for five weeks according to the NRC (1994) with the broiler chick starter feed obtained from a commercial company. The feed was characterized by 23% content of crude protein (CP) and 3100 kcal/kg of metabolic energy (ME), 0.92 g/kg of calcium, and 0.47 g/kg of phosphorus supplemented without or with black cumin powder BCP. From days 1 to 35, the quail birds were fed experimental diets formed by mixing BCP at levels of 1%, 2%, and 4% to the feed, and a diet without BCP as a control group. A nipple drinking device was used to deliver freshwater for 24 h.

### 2.3. Arrangement of Experiments

At the start of the experiment, 240 quail chicks were weighed using an electrical balance with an accuracy of 0.01 g, and the mean live weight in each group was uniform. The study included four groups of 15 chicks each, with each group having 1, 2, or 4% BCP, and a control without BCP. The thermostat radiator was used to set the temperature at 33 °C for the first week, and afterward, the temperature was gradually reduced by 2–3 °C each week until it was set at 24–25 °C. With the air conditioner, the room temperature was set to a comfortable level for the quails, and the temperature was also monitored with a thermometer. The quail experiment was maintained for an estimated 35 days. Feed and water were accessible at any time, and natural and artificial illumination was used for a total of 24 h.

### 2.4. Determination of Growth Performance and Carcass Characteristics

The performance of the birds including growth parameters such as the body weight (BW), body weight gain (BWG) feed intake (FI), and feed conversion ratio (FCR = g feed/g gain) were measured on a weekly basis. At 5-weeks-old, 32 quails were used for the carcass examinations. Following the weight of the hot carcass, the weight of the belly fat and inner organs such as the heart, liver, and gizzard were measured. To study the cold carcass weight, the carcasses were maintained at +4 °C for 24 h.

### 2.5. Determination of Shelf Life in Breast Meat Samples

At the conclusion of the study, the samples were kept at 4 °C for 0, 3, 5, and 7 days, with various impacts on the meat quality being studied. The breast meat samples from each subgroup of quails were obtained, which were the sliced and the carcass traits assessed.

### 2.6. Oxidation Analysis

This oxidation analysis included three parameters: Peroxide value analysis, thiobarbituric acid number (TBARS), and microbial analysis (total bacterial count, TBC).

To assess the oxidation state in the flesh from the samples from the quails maintained at 4 °C for 0, 3, 5, and 7 days, peroxide content analysis was conducted using the AOAC 965.33 technique [32].

The lipid oxidation status of two breast meat samples collected from each group stored at 4 °C, after 0, 3, 5, and 7 days of preservation, was determined by the thiobarbituric acid number analysis (TBA).

The total psychrophilic counts of the quail breast flesh samples were conducted at 0, 3, 5, and 7 days. For this, 10 g of the quail breast flesh samples was homogenized for 1 min in 90 mL of 0.1 percent peptone water. Diluting the homogeneous mixture with 0.1 percent peptone water and percent ringer solution yielded serial dilutions. In the total psychrophilic live count study, the plate count agar (PCA) medium was employed. For 7 days, Petri dishes were incubated at 7 °C. Log CFU/g is the unit of measurement for the number of microbiological bacteria.

### 2.7. Determination of pH in Meat

Two samples of breast flesh were obtained from each subgroup at the end of the experiment, and the pH level of the breast flesh was tested at 0, 3, 5, and 7 days. On day 1, a Testo 205 meat and food pH meter was used to detect the pH level. Measurements were taken from three separate regions of the breast flesh for this goal, and the mean of these results was computed. The breast flesh was processed through a blender and combined with purified water after taking a 5 g sample and homogenized to measure the pH value. The resulting homogeneous liquid was filtered, and the pH of the breast flesh was determined using a pH meter with a probe [33].

### 2.8. Color Measurement

Following the experiment, the color of the quail breast and thigh flesh was measured (from two separate regions) using a Konica Minolta Chromometer model CR-300 colorimeter equipment. Meat values were assessed using a chronometer (L* measures comparative lightness, a* measures comparative redness, and b* measures comparative yellowness) [33]. Calibration was conducted with the black and white plates prior to actually beginning the measurement.

### 2.9. Sensory Evaluation

On the first day, a 10-member panel assessed the sensory quality of the quail thigh and breast flesh. The Niğde Ömer Halisdemir University faculty staff and students were among the panelists. Each sample was covered in aluminum foil and cooked for 75 min at 175 °C [34]. Panelists were offered the breast meat items on enclosed serving plates labeled with 3-digit random numeric values.

### 2.10. Statistical Analysis

The data were evaluated as a completely randomized design with four treatments, and statistical analysis was performed using the SPSS 11.0 general linear model techniques (SPSS Inc., Chicago, IL, USA). Significant differences between the groups were confirmed by the Duncan’s multiple range test. Differences with *p* < 0.05 were considered as significant (a, b) and *p* < 0.01 as highly significant (A, B).

## 3. Results

The obtained results demonstrated that the addition of BCP had no effect on the daily feed intake during all five weeks of the experiment. However, at the fourth week of the experiment, significant differences in body weight gain were noticed (*p* = 0.029). In a group fed with a 2% BCP diet, the highest body weight gain was noted. At the third week of the experiment, the feed conversion ratio was statistically different (*p* = 0.050). The highest value of the feed conversion ratio was noted in the 2% BCP group (Table 1).

During the experiment, all of the birds remained healthy, and no deaths were documented. Significant differences in the mean live BW body weight between the treatments were noticed since third week of the experiment (Table 2). Moreover, at the fifth week of the experiment, significant differences in the female live BW body weight were noted between the groups (*p* = 0.046).

Table 3 shows the impact of the experimental diets on the carcass characteristics of the quails. The dietary treatments had no effect on the live weight, cold carcass weight, and carcass efficiency. However, significant differences were noticed in the mean hot carcass weight between groups (*p* = 0.039).

Table 4 shows the effects of different levels of BCP supplementation on the commercial cut of the quail carcass. Significant differences were noted in the mean percentage share of the breast and neck (*p* < 0.01) and thigh (*p* < 0.05) in the quail carcasses. Moreover, the statistical differences were noted in the male percentage share of the breast (*p* = 0.008) and neck (*p* = 0.026). The statistical differences were also noted in the cut of the female quail carcass such as percentage share of the thigh (*p* = 0.050) and wings (*p* = 0.039).

No statistical differences were observed in the giblet proportion and abdominal fat between the different levels of BCP supplementation (Table 5).

Table 6 shows the changes in the peroxide values in the breast meat after 0, 3, 5, and 7 days of storage at 4 °C in the refrigerator. In all tested periods, the peroxide value of the breast flesh dropped when the BCP amount was increased. The control group had the greatest peroxide value compared to all of the experimental dietary groups, on all days. Compared to the other groups, the 4% BCP supplemented group had the lowest peroxide value in the breast flesh.

Along with increasing storage duration up to day 7, the quantity of psychrophilic bacteria in all samples increased continuously. At days 0, 5, and 7, the total number of psychrophilic bacteria in the breast flesh revealed a statistically significant difference (*p* < 0.01) between the groups. The overall number of psychrophilic bacteria in each group grew as the storage duration increased. On the other hand, the overall number of psychrophilic bacteria in each group decreased as the BCP inclusion increased.

Table 6 shows the impact of storage on the pH of the breast flesh. The pH value of the breast flesh discovered increased with the storage time, peaking at day 7. The BCP-supplemented groups (1%, 2% and 4%) were characterized by statistically lower (*p* = 0.000) pH values compared to the control group.

The TBA levels in the breast meat were measured after storage at 4 °C for 0, 3, 5, and 7 days, which are summarized in Table 6. The control group had the greatest TBA value compared to the other groups. The diet supplemented by the 4% BCP level had the lowest breast flesh TBA value (*p* = 0.000). The TBA value of the breast meat decreased as the level of BCP in the diet increased. The antioxidant capacity of the BCP utilized in the study was efficient in avoiding oxidation in the quail breast flesh.

Moreover, the quail thigh and breast color such as L* (brightness), a* (redness), and b* (yellowness) were examined in the cold carcass (Table 7). In terms of the L* of the thigh meat and the b* value of the thigh skin in the cold carcass, there were significant differences (*p* < 0.05) between the treatments. Table 7 also shows the findings of the color L* (brightness), a* (redness), and b* (yellowness) values of the quail breast meat and skin. In terms of the L*, a*, and b* values in the breast flesh, there were no statistically significant differences between the groups. However, in the breast skin, significant differences were noticed in the L* (*p* = 0.002) and b* (*p* = 0.025) value between groups.

The effects of the dietary BCP supplementation on the perceived quail meat breast and thigh ratings are shown in Table 8. The thigh obtained from the experimental groups had higher values of color (*p* = 0.000), juiciness (*p* = 0.000), tenderness (*p* = 0.001), and flavor (*p* = 0.002) compared to the control group. While the breast obtained from the experimental groups had statistically higher values of color (*p* = 0.006), juiciness (*p* = 0.050), and flavor (*p* = 0.001) compared to the control group.

The effects of dietary BCP supplementation on the chemical composition of the quail meat breast and thigh results are shown in Table 9. The experimental groups had substantially greater values of DM and CP of the thigh and breast meat than the control group.

## 4. Discussion

The current study offers information on the effects of using different levels of BCP supplementation on the growth performance, carcass characteristics, the effect on lipid peroxidation during storage, microbial load, TBA values, pH, color, and the sensory characteristics of the meat of the quails. The obtained data showed that adding BCP to the quail diets had no effect on weight gain during the first three weeks. However, BCP supplementation had a significant increase in weight gain in the last two weeks, thus affecting the feed conversion ratio. The current findings support those of El-Hack et al. [35] and Salam et al. [36], who found that feeding quail diets supplemented with BCP increased the digestive enzyme production, improving the nutrient digestibility and growth performance. The content of volatile oil or essential oil has biological activities that might work not only as anti-bacterials and antioxidants, but also as a stimulant of digestive enzymes in the intestinal mucosa and pancreas, therefore improving the dietary nutrient digestion [37]. Our finding contradicts the data obtained by Abbas and Ahmed [38], who found that the birds fed a meal enriched with 1% or 2% black cumin had considerably reduced BWG and unchanged FCR. However, Naula et al. [39] showed an impact of black cumin additive on the BWG, feed intake, FCR, and weight of different body organs (breast and thigh). Seidavi et al. [40] reported that supplementing quail diets with BCP oil (2% and 0.5%) enhanced the performance (growth and egg production) while also reducing the pathogenic bacteria in the gut. Our own data indicated that birds given 2% BCP had the highest feed intake and weight growth. Similarly, El-Hack [35] reported that there was a significant difference between the BCP supplemented groups in comparison to the control. However, Emam et al. [41] reported that there was no significant influence of the BWG and FCR on the layers by the dietary supplementation of BCP in layers.

The findings in the current study on the quail carcass yield were consistent with prior studies on the effects of BCP or its extract. Schemmer et al. [42], Tufan et al. [43], and Khan et al. [44] reported that BCP supplementation had no effect on the dressing percentage values of the quails and broilers. According to Kumar et al. [45], quails given a feed enhanced with BCP had larger breast and thigh weights as well as a higher dressing percentage compared to the control.

These findings support the findings of Khan et al. [44], who found that the value of the dressing percent was enhanced in the highest fed BCP addition group. Majeed et al. [46] reported that their research findings showed no improvement in the gizzard or liver after using BCP in the diet. Furthermore, Jahan et al. [47] reported that the inclusion of the BCP extract (100–300 mg/kg) had no significant effect on the dressing percentage. Similarly, Seidavi et al. [40] reported that dietary factors had no effect on the gizzard and dressing percentage.

The experimental diet considerably reduced the gizzard weight of the birds at the end of the feeding trial in our investigation, but had no effect on the rest of the internal organs. Detectable anomalies in the internal organ weights are a sensitive indicator for reducing anti-nutritional factors following hazardous chemical exposure. Hassanien et al. [48], Sogut et al. [49], and Ismail [50] reported that there were no significant effects of the dietary BCS or BCS extract on the dressing percentage, edible inner organs, or chicken belly fat.

The 4% BCP supplemented group noted the lowest peroxide and the TBA values in the breast flesh compared to the control group. The results indicated that the BCP antioxidant capacity was efficient in avoiding oxidation in the quail breast flesh. This suggests the utility of the peroxide value as a quick and sensitive method for detecting changes in quail meat storage deterioration. These findings are consistent with those reported by Kumar et al. [45], Guler et al. [51], and Zwolan et al. [52]. BCP has been demonstrated in the literature to extend the shelf life of refrigerated meat, which might be related to the presence of flavonoid and phenolic chemicals, which help to inhibit lipid oxidation. BCP possesses antibacterial, antioxidant, and anti-inflammatory properties, which may contribute to its positive benefits on immunity and growth performance [53,54].

In our study, it was observed that dietary inclusion of BCP increases in the quails’ diet caused a reduction in the pH continuously when compared to the control group. Chandralekha et al. [55] also reported that the increase in pH following storing was considerably lower in all of the pomegranate rind extract treated groups than in the control group. This conclusion is consistent with the data reported by Rahman and Kim [56], who found that the sensory meat characteristics and pH values of the breast meat were substantially reduced in different *Nigella sativa* treatments when compared to the control.

The sensory evaluations of the control thigh meat were poorer than those of the experimental treatments. The differences in the tenderness of the breast flesh between the treatments and control group were determined to be non-significant (*p* > 0.05). Color, juiciness, and flavor of the breast meat were observed to be significantly different between the control group and the experimental, BCP augmented groups. Color, juiciness, and flavor numerical values were lower in the control group than 1%, 2%, and 4% BCP supplemented groups. Similarly, Adegbeye et al. [57] and Singh et al. [58] reported that dietary regimens substantially impacted the sensory characteristics investigated, with the exception of meat juiciness.

## 5. Conclusions

Supplementation of BCP is considered to be safe because no acute hazardous adverse effects were detected over the testing period. The experimental diets reduced the gizzard weight and belly fat, but had no influence on the quail’s growth performance or carcass attributes, according to our findings. It may also be concluded that dietary supplementation with BCP had an effect on the quail’s immunological response, antibacterial effects most likely due to BCP’s significant antioxidant activity. However, the supplementation of BCP in the feed boosts the natural antibody production and reduces the psychrophilic bacteria. The breast meat pH, thiobarbituric acid, and peroxide values were determined to be lower as the quantity of BCP increased. These findings revealed that BCP contains antioxidant-active phenolic compounds that are efficient in preventing or delaying meat oxidation and may be utilized as a natural antioxidant in quail feeding.

As a result, considering all the parameters studied in the study, it was concluded that BCP had a significant effect on the quail performance and meat shelf life and could be used in poultry mixed feed to prevent or delay the lipid oxidation of meat.

## Figures and Tables

**Table 1 animals-12-01298-t001:** The effect of BCP supplementation at different levels on the weekly body weight gain (g), feed intake (g), and feed conversion ratio.

Groups		Control	1% BCP	2% BCP	4% BCP	SEM	*p*
Body weight gain (BWG)							
Weeks	1	32.28 ± 0.61	32.15 ± 0.52	32.64 ± 0.61	32.47 ± 0.81	0.639	0.953
	2	60.98 ± 1.71	58.55 ± 0.37	60.70 ± 2.31	61.25 ± 2.06	1.621	0.703
	3	79.75 ± 0.44	79.02 ± 0.42	77.31 ± 2.43	77.95 ± 1.21	1.123	0.618
	4	66.60 ± 1.01 ^ab^	62.77 ± 3.83 ^ab^	70.03 ± 3.11 ^a^	65.54 ± 0.92 ^b^	2.224	0.029
	5	39.90 ±1.86	42.43 ± 2.56	44.69 ± 2.24	36.11 ± 2.37	2.255	0.058
Feed intake (FI)							
Weeks	1	47.23 ± 1.10	46.27 ± 0.44	45.61 ± 0.62	47.00 ± 1.52	0.918	0.672
	2	105.84 ± 3.34	102.16 ± 1.23	105.93 ± 3.53	105.89 ± 3.35	2.860	0.767
	3	157.25 ± 4.14	158.11 ± 2.17	169.16 ± 12.37	157.25 ± 2.33	5.257	0.542
	4	213.59 ± 6.69	207.32 ± 7.08	207.30 ± 3.65	208.62 ± 3.58	5.254	0.560
	5	228.73 ± 5.92	234.02 ± 7.52	232.77 ± 4.17	220.25 ± 7.22	6.213	0.443
Feed conversion ratio (FCR)							
Weeks	1	1.46 ± 0.01	1.44 ± 0.01	1.39 ± 0.03	1.44 ± 0.04	0.027	0.487
	2	1.73 ± 0.01	1.74 ± 0.01	1.75 ± 0.03	1.73 ± 0.01	0.020	0.509
	3	1.97 ± 0.05 a	2.00 ± 0.01 ^a^	2.20 ± 0.19 ^b^	2.01 ± 0.03 ^a^	0.033	0.050
	4	3.21 ± 0.13	3.33 ± 0.14	3.98 ± 0.14	3.18 ± 0.07	0.120	0.300
	5	5.77 ± 0.32	5.57 ± 0.36	5.24 ± 0.23	6.16 ± 0.36	0.324	0.300

SEM: standard mean error, BCP: black cumin powder; ^a,b^—significant differences at the level of *p* < 0.05.

**Table 2 animals-12-01298-t002:** The effect of BCP supplementation at different levels on the weekly live body weight (LBW).

Weeks		Groups				SEM	*p*
		Control	1% BCP	2% BCP	4% BCP		
DOC		9.08 ± 0.11	9.10 ± 0.12	9.20 ± 0.14	9.21 ± 0.12	0.125	0.566
1		41.36 ± 0.81	41.25 ± 0.61	41.83 ± 0.69	41.67 ± 0.84	0.739	0.160
2	M	102.83 ± 2.10	100.93 ± 1.58	105.75 ± 1.96	103.92 ± 2.28	1.981	0.418
	F	101.72 ± 2.78	98.32 ± 2.14	99.60 ± 2.16	101.49 ± 2.08	2.295	0.560
	Mean	102.35 ± 1.67	99.81 ± 1.28	102.54 ± 1.52	102.92 ± 1.59	1.520	0.460
3	M	185.97 ± 4.58	178.39 ± 2.77	185.13 ± 3.63	180.88 ± 3.44	3.609	0.349
	F	179.37 ± 2.60	179.23 ± 2.64	176.03 ± 5.11	180.87 ± 2.74	3.277	0.700
	Mean	182.20 ± 2.48 ^A^	178.83 ± 1.89 ^B^	179.84 ± 3.37 ^AB^	180.87 ± 2.15 ^AB^	2.477	0.001
4	M	252.72 ± 4.91	242.45 ± 3.75	249.05 ± 4.69	244.86 ± 4.52	4.471	0.376
	F	245.48 ± 3.15	240.83 ± 3.52	250.18 ± 3.63	247.83 ± 3.61	3.482	0.274
	Mean	248.75 ± 2.83 ^AB^	241.60 ± 2.54 ^B^	249.72 ± 2.85 ^A^	246.41 ± 2.84 ^AB^	2.767	0.016
5	M	286.75 ± 5.73	288.21 ± 4.34	291.46 ± 3.52	282.32 ± 6.10	4.927	0.086
	F	289.96 ± 5.30 ^ab^	280.21 ± 4.12 ^b^	296.17 ± 5.10 ^a^	282.67 ± 3.82 ^b^	4.587	0.046
	Mean	288.51 ± 3.85 ^ab^	284.03 ± 3.01 ^b^	294.26 ± 3.34 ^a^	282.52 ± 3.37 ^b^	3.396	0.050

SEM: standard mean error, BCP: black cumin powder, M: male, F: female, DOC: day old chick; ^A,B^—highly significant differences at the level of *p* < 0.01; ^a,b^—significant differences at the level of *p* < 0.05.

**Table 3 animals-12-01298-t003:** The effect of BCP supplementation at different levels on the carcass characteristics.

Groups		Live Weight (g)	Hot Carcass Weight (g)	Cold Carcass Weight (g)	Carcass Efficiency (%)
Control	Mean	283.21 ± 2.98	213.04 ± 2.07 ^b^	214.65 ± 1.95	75.81 ± 0.31
1% BCP		283.28 ± 3.01	214.75 ± 2.74 ^ab^	213.93 ± 2.94	75.50 ± 0.45
2% BCP		292.04 ± 3.61	221.66 ± 2.83 ^a^	221.91 ± 2.34	76.02 ± 0.41
4% BCP		289.01 ± 4.19	219.17 ± 3.56 ^ab^	219.61 ± 3.38	75.99 ± 0.53
SEM		1.917	1.202	1.187	0.306
*p*		0.660	0.039	0.991	0.657
Control	M	276.51 ± 3.35	209.24 ± 2.96	212.07 ± 2.88	76.69 ± 0.32
1% BCP		275.16 ± 2.63	210.75 ± 3.00	208.59 ± 3.55	75.77 ± 0.74
2% BCP		285.00 ± 3.63	215.48 ± 2.48	217.53 ± 2.46	76.33 ± 0.31
4% BCP		286.43 ± 7.16	219.25 ± 5.82	219.11 ± 5.71	76.50 ± 0.62
SEM		4.191	3.567	3.653	0.493
*p*		0.210	0.251	0.214	0.650
Control	F	289.91 ± 3.75	216.84 ± 2.35	217.23 ± 2.48	74.94 ± 0.32
1% BCP		291.39 ± 3.65	218.75 ± 4.30	219.28 ± 4.02	75.22 ± 0.55
2% BCP		299.08 ± 5.35	227.85 ± 4.15	226.29 ± 3.44	75.71 ± 0.78
4% BCP		291.60 ± 4.69	219.09 ± 4.54	220.10 ± 4.04	75.49 ± 0.86
SEM		1.457	1.398	1.466	0.436
*p*		0.360	0.855	0.920	0.629

SEM: standard mean error, M = Male, F = Female, BCP: black cumin powder. ^a,b^—significant differences at the level of *p* < 0.05.

**Table 4 animals-12-01298-t004:** The effect of the different levels of BCP supplementation on the commercial cut of the quail carcass (%).

Groups		Thigh (%)	Breast (%)	Back (%)	Wings (%)	Neck (%)
Control	Mean	32.98 ± 0.23 ^a^	34.22 ± 0.44 ^B^	14.39 ± 0.41	8.91 ± 0.14	6.73 ± 0.19 ^A^
1% BCP		32.21 ± 0.47 ^b^	35.29 ± 0.38 ^AB^	14.38 ± 0.39	9.17 ± 0.22	6.44 ± 0.26 ^AB^
2% BCP		32.04 ± 0.22 ^b^	34.90 ± 0.38 ^B^	14.47 ± 0.25	9.29 ± 0.17	6.52 ± 0.13 ^AB^
4 % BCP		31.79 ± 0.26 ^ab^	36.36 ± 0.43 ^A^	14.14 ± 0.26	9.09 ± 0.16	6.05 ± 0.17 ^B^
SEM		0.295	0.409	0.327	0.172	0.185
*p*		0.050	0.004	0.908	0.475	0.010
Control	M	32.71 ± 0.39	34.17 ± 0.58 ^AB^	14.08 ± 0.56	8.79 ± 0.23	7.05 ± 0.27 ^a^
1% BCP	M	32.62 ± 0.66	34.86 ± 0.54 ^B^	14.25 ± 0.62	9.35 ± 0.32	6.98 ± 0.37 ^a^
2% BCP	M	32.25 ± 0.19	34.08 ± 0.63 ^AB^	14.75 ± 0.43	9.22 ± 0.31	6.70 ± 0.15 ^ab^
4% BCP	M	31.93 ± 0.28	36.73 ± 0.51 ^A^	13.77 ± 0.27	9.04 ± 0.21	5.96 ± 0.22 ^b^
SEM		0.376	0.560	0.472	0.270	0.252
*p*		0.536	0.008	0.564	0.524	0.026
Control	F	33.24 ± 0.23 ^a^	34.28 ± 0.71	14.69 ± 0.61	9.02 ± 0.16 ^b^	6.41 ± 0.22
1% BCP	F	31.80 ± 0.69 ^b^	35.74 ± 0.53	14.52 ± 0.50	8.98 ± 0.31 ^ab^	5.90 ± 0.26
2% BCP	F	31.83 ± 0.39 ^b^	35.73 ± 0.22	14.18 ± 0.27	9.36 ± 0.15 ^a^	6.34 ± 0.20
4% BCP	F	31.64 ± 0.46 ^b^	35.99 ± 0.71	14.50 ± 0.41	9.13 ± 0.25 ^a^	7.16 ± 0.26
SEM		0.446	0.326	0.235	0.140	0.380
*p*		0.050	0.069	0.519	0.039	0.144

SEM: standard mean error, M: Male, F: Female, BCP: black cumin powder; ^A,B^—highly significant differences at the level of *p* < 0.01; ^a,b^—significant differences at the level of *p* < 0.05.

**Table 5 animals-12-01298-t005:** The effect of the different levels of BCP supplementation on giblet proportion and abdominal fat (%).

Groups		Heart (%)	Liver (%)	Gizzard (%)	Abdominal Fat (%)
Control	Mean	1.24 ± 0.05	3.03 ± 0.11	2.55 ± 0.13	2.06 ± 0.16
1% BCP	Mean	1.22 ± 0.04	3.28 ± 0.17	2.41 ± 0.06	1.91 ± 0.15
1% BCP	Mean	1.21 ± 0.04	3.09 ± 0.14	2.26 ± 0.07	2.05 ± 0.07
4% BCP	Mean	1.23 ± 0.03	3.29 ± 0.13	2.38 ± 0.09	2.06 ± 0.14
SEM		0.045	0.045	0.086	0.134
*p*		0.986	0.986	0.180	0.843
Control	M	1.25 ± 0.08	2.74 ± 0.10	2.37 ± 0.22	2.19 ± 0.22
1% BCP	M	1.33 ± 0.06	3.22 ± 0.23	2.38 ± 0.10	2.17 ± 0.25
2% BCP	M	1.22 ± 0.05	3.92 ± 0.23	2.16 ± 0.07	2.05 ± 0.13
4% BCP	M	1.23 ± 0.04	3.09 ± 0.10	2.33 ± 0.12	2.10 ± 0.19
SEM		0.059	0.167	0.127	0.197
*p*		0.615	0.272	0.656	0.957
Control	F	1.22 ± 0.07	3.32 ± 0.12	2.72 ± 0.11	1.93 ± 0.23
1% BCP	F	1.10 ± 0.04	3.34 ± 0.27	2.45 ± 0.08	1.65 ± 0.16
2% BCP	F	1.21 ± 0.07	3.26 ± 0.15	2.37 ± 0.09	2.05 ± 0.08
4% BCP	F	1.22 ± 0.06	3.50 ± 0.22	2.43 ± 0.14	2.06 ± 0.24
SEM		0.064	0.190	0.109	0.176
*p*		0.515	0.849	0.140	0.448

SEM: standard mean error, *p*: Significance, M: Male, F: Female, BCP: black cumin powder.

**Table 6 animals-12-01298-t006:** The effect of the different levels of BCP supplementation on the breast meat peroxide value, total psychrophilic bacteria count, the effect of storage on breast meat pH values, and the breast meat thiobarbituric acid (TBA) value (mg MDA/kg).

Groups		Control	1% BCP	2% BCP	4% BCP	SEM	*p*
Breast Meat Peroxide Value (meq/kg)							
Days	0	2.00 ± 0.00 ^A^	1.00 ± 0.00 ^B^	1.00 ± 0.00 ^B^	0.50 ± 0.50 ^B^	0.125	0.000
	3	3.50 ± 0.50 ^A^	2.49 ± 0.49 ^AB^	1.99 ± 0.00 ^BC^	1.00 ± 0.00 ^C^	0.248	0.000
	5	4.47 ± 0.50 ^A^	3.50 ± 0.50 ^AB^	2.50 ± 0.50 ^B^	2.00 ± 0.00 ^B^	0.377	0.000
	7	6.00 ± 0.00 ^A^	4.50 ± 0.49 ^AB^	3.48 ± 0.50 ^BC^	2.50 ± 0.49 ^C^	0.375	0.010
Breast Meat Total Psychrophilic Bacteria Count (log CFU/g)							
Days	0	2.10 ± 0.03 ^A^	1.18 ± 0.03 ^B^	0.95 ± 0.05 ^C^	0.74 ± 0.04 ^D^	0.035	0.000
	3	2.29 ± 0.02	2.12 ± 0.03	2.04 ± 0.50	1.58 ± 0.01	0.135	0.343
	5	2.64 ± 0.01 ^A^	2.44 ± 0.02 ^B^	2.39 ± 0.01 ^BC^	2.35 ± 0.04 ^C^	0.016	0.001
	7	3.65 ± 0.04 ^A^	3.33 ± 0.01 ^B^	3.27 ± 0.01 ^B^	3.23 ± 0.03 ^B^	0.022	0.001
The Effect of Storage on Breast Meat pH Values							
Days	0	5.95 ± 0.02 ^A^	5.89 ± 0.01 ^B^	5.78 ± 0.02 ^C^	5.62 ± 0.02 ^D^	0.015	0.000
	3	6.01 ± 0.00 ^A^	5.92 ± 0.03 ^B^	5.83 ± 0.02 ^C^	5.72 ± 0.00 ^D^	0.015	0.000
	5	6.10 ± 0.04 ^A^	5.94 ± 0.03 ^B^	5.85 ± 0.01 ^C^	5.73 ± 0.00 ^D^	0.019	0.000
	7	6.13 ± 0.00 ^A^	5.96 ± 0.03 ^B^	5.88 ± 0.00 ^C^	5.77 ± 0.01 ^D^	0.010	0.000
Breast Meat Thiobarbituric Acid (TBA) Value (mg MDA/kg)							
Days	0	0.26 ± 0.02 ^A^	0.25 ± 0.00 ^AB^	0.22 ± 0.00 ^AB^	0.21 ± 0.01 ^B^	0.010	0.000
	3	0.36 ± 0.00 ^A^	0.33 ± 0.00 ^B^	0.31 ± 0.00 ^B^	0.28 ± 0.01 ^C^	0.000	0.000
	5	0.41 ± 0.00 ^A^	0.36 ± 0.00 ^B^	0.35 ± 0.00 ^B^	0.34 ± 0.00 ^B^	0.000	0.000
	7	0.55 ± 0.00 ^A^	0.49 ± 0.01 ^B^	0.38 ± 0.01 ^C^	0.37 ± 0.00 ^C^	0.000	0.000

SEM: standard mean error, BCP: black cumin powder; ^A,B,C,D^—highly significant differences at the level of *p* < 0.01.

**Table 7 animals-12-01298-t007:** The effect of the different levels of BCP supplementation on the thigh meat, thigh skin, breast meat, and breast skin in cold carcass.

Groups	Thigh Meat			Thigh Skin			Breast Meat			Breast Skin		
	L*	a*	b*	L*	a*	b*	L*	a*	b*	L*	a*	b*
Control	53.76 ± 0.66 ^ab^	2.18 ± 0.35	6.46 ± 0.42	58.57 ± 1.25	1.96 ± 0.28	6.36 ± 0.47 ^a^	62.31 ± 0.67	4.01 ± 0.38	10.74 ± 0.34	68.06 ± 0.42 ^A^	3.75 ± 0.30	9.79 ± 0.46 ^b^
1% BCP	56.08 ± 0.57 ^a^	2.33 ± 0.39	7.14 ± 0.46	57.90 ± 0.68	1.65 ± 0.19	5.61 ± 0.31 ^b^	60.35 ± 0.80	3.67 ± 0.37	10.21 ± 0.30	68.69 ± 0.66 ^A^	3.83 ± 0.54	12.46 ± 1.20 ^a^
2% BCP	55.48 ± 0.56 ^b^	1.33 ± 0.23	6.48 ± 0.34	57.27 ± 0.99	1.13 ± 0.20	5.06 ± 0.32 ^b^	62.11 ± 1.10	3.65 ± 0.31	10.68 ± 0.22	68.59 ± 0.55 ^A^	3.34 ± 0.23	9.86 ± 0.25 ^b^
4% BCP	55.89 ± 0.39 ^b^	1.76 ± 0.31	6.31 ± 0.27	57.01 ± 0.84	1.67 ± 0.41	4.82 ± 0.49 ^ab^	60.60 ± 0.77	3.75 ± 0.29	10.55 ± 0.29	65.51 ± 0.83 ^B^	3.80 ± 3.33	9.60 ± 0.48 ^b^
SEM	0.551	0.327	0.377	0.945	0.271	0.404	0.733	0.436	0.444	0.752	0.247	0.547
*p*	0.019	0.154	0.426	0.670	0.227	0.050	0.395	0.077	0.793	0.002	0.347	0.025

SEM: standard mean error, M: Male, F: Female, BCP: black cumin powder; L* measures the relative lightness, a* the relative redness, and b* the relative yellowness; ^A,B^—highly significant differences at the level of *p* < 0.01; ^a,b^—significant differences at the level of *p* < 0.05.

**Table 8 animals-12-01298-t008:** The sensory characteristics of the cooked meat of the Japanese quail as influenced by dietary BCP.

Parameters	Control	1% BCP	2% BCP	4% BCP	SEM	*p*-Value
Color						
Thigh	6.68 ± 0.33 ^C^	7.31 ± 0.20 ^B^	8.43 ± 0.16 ^A^	7.81 ± 0.14 ^B^	0.130	0.000
Breast	6.75 ± 0.42 ^B^	7.12 ± 0.38 ^B^	8.37 ± 0.18 ^A^	7.50 ± 0.17 ^AB^	0.180	0.006
Juiciness						
Thigh	6.86 ± 0.34 ^C^	7.43 ± 0.23 ^BC^	8.50 ± 0.11 ^A^	8.00 ± 0.17 ^AB^	0.138	0.000
Breast	6.24 ± 0.23 ^b^	6.75 ± 0.50 ^ab^	7.87 ± 0.33 ^a^	6.51 ± 0.51 ^b^	0.200	0.050
Tenderness						
Thigh	6.92 ± 0.28 ^C^	7.43 ± 0.19 ^BC^	8.18 ± 0.17 ^A^	8.07 ± 0.16 ^AB^	0.124	0.001
Breast	6.76 ± 0.45	6.75 ± 0.48	7.87 ± 0.26	7.50 ± 0.24	0.200	0.123
Flavor						
Thigh	6.82 ± 0.27 ^B^	7.51 ± 0.27 ^A^	8.07 ± 0.14 ^A^	7.51 ± 0.14 ^A^	0.115	0.002
Breast	6.14 ± 0.29 ^B^	6.38 ± 0.53 ^B^	8.51 ± 0.26 ^A^	6.82 ± 0.35 ^B^	0.240	0.001

SEM: standard mean error, BCP: black cumin powder; ^A,B,C^—highly significant differences at the level of *p* < 0.01; ^a,b^—significant differences at the level of *p* < 0.05.

**Table 9 animals-12-01298-t009:** The effect of BCP supplementation on the chemical composition (proximate %) of the thigh and breast meat of the quail birds.

Parameters	Control	1% BCP	2% BCP	4% BCP	SEM	*p*-Value
Dry Matter (DM)						
Thigh	31.46 ± 0.00 ^a^	31.81 ± 0.50 ^a^	29.75 ± 0.92 ^b^	30.97 ± 0.00 ^b^	0.173	0.044
Breast	32.48 ± 0.63 ^b^	34.51 ± 1.01 ^ab^	36.26 ± 0.29 ^a^	37.80 ± 1.27 ^a^	0.204	0.048
Crude Protein (CP)						
Thigh	14.56 ± 0.38 ^ab^	13.92 ± 0.04 ^b^	15.11 ± 0.09 ^a^	14.96 ± 0.39 ^ab^	0.128	0.012
Breast	14.60 ± 0.05 ^C^	18.24 ± 0.15 ^A^	16.39 ± 0.11 ^B^	14.75 ± 0.01 ^C^	0.084	0.000
Ether Extract (EE)						
Thigh	2.16 ± 0.01	2.28 ± 0.03	2.36 ± 0.04	2.29 ± 0.17	0.066	0.267
Breast	2.06 ± 0.01	2.15 ± 0.05	2.19 ± 0.06	2.17 ± 0.08	0.055	0.463
Crude Ash (CA)						
Thigh	0.75 ± 0.09	0.89 ± 0.05	0.94 ± 0.22	0.85 ± 0.10	0.110	0.704
Breast	1.76 ± 0.21	1.64 ± 0.09	1.59 ± 0.03	1.48 ± 0.11	0.113	0.053

SEM: standard mean error, BCP: black cumin powder; ^A,B,C^—highly significant differences at the level of *p* < 0.01; ^a,b^—significant differences at the level of *p* < 0.05.

## Data Availability

The data presented in this study are available on request from the corresponding author.

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
