# Peer review of "Effect of Dietary Supplementation of Black Cumin Seeds (Nigella sativa) on Performance, Carcass Traits, and Meat Quality of Japanese Quails (Coturnix coturnix japonica)"

_animals, 2022, doi:10.3390/ani12101298_

Round 1

Reviewer 1 Report

Pls see the attached file.

REGARDS

Author Response

The authors would like to thank the reviewers for all comments and suggestions which may improve manuscript quality. The authors made changes to the text in line with the reviewers' renewed comments.

Reviewer 1

Reviewer 2 Report

The manuscript deals with a very topical issue: the attempt to reduce the use of antibiotics in livestock farming. This is not least against the background of the increasing occurrence of therapy-resistant germs in the human sector, which is attributed to antibiotic abuse in animal husbandry. The authors should have written one or two sentences on this. Instead, the emphasis is on the function of antibiotics as growth promoters. However, these have already been banned for many years, so that it cannot be the aim of this work to find a replacement. Instead, the focus should be more on the aspect of intestinal health.

The experimental design is clearly structured and has been implemented with sufficient precision. In the results section, however, there should not only be a juxtaposition of tables, but Tet should be inserted between the tables to summarise the most important results.

In the reproduction of the data, dimensions are sometimes missing. For example, it is not clear whether the BWG are given in grams or kilograms. It is of course understandable that it can only be grams, but for the sake of order, the dimension should be given here.

The English language also needs some revision here and there.

In addition to these general remarks, some specific indications are necessary:

Line 18            germs? Or do you mean bacteria?

Line 21            Nigella sativa please cursive

Line 25            supplemented group? Not the group was supplemented; group with 2% BCP diet

Line 31             especially feeding the 2% BCP diet

Line 46            growth promoters have not been allowed for a long time; nowadays it is a matter of reducing the use of antibiotics in livestock farming, especially in view of the resistant germs in the human sector.

Line 47            Escherichia coli

Line 63            Because of the fat deposition or storage

Line 76            e.g.

Line 93            delete the and before 1,3,8-p-acetic

Line 97            which have effects against fungi, bacteria, spasmodic, inflammation and analgesic effects as well

Line 111          NOHU ? first explain, than the abbreviation can be used

Line 113          were allocated into

Line 114          Mixed sex groups were put

Line 115          birds at the beginning was kept

Line 117          delete the comma behind weeks

Line 118          The diet contains 23% crude protein and 3100 Kcal metabolic energie/kg as fed, 0.92 g Ca/kg, 0.47 g P/kg and was supplemented without or with (without is the control, therefore it should be listed at first)

Line 127          1, 2 or 4%

Line 129          33 °C

Line 130          dito (use always a blank between the number and °C; also in the following text)

Line 174          Chronometer or chromometer

Line 196          second line in the table: control and 1% BCP are fat, 2 and 4% not

Line 206          table 2 should be table 1: first the feed intake than body weight gain (as it results from the feed intake)

Line 206          in the table: what is the dimension of BWG? g? kg? And feed intake (also gram?)

Line 218          Table 4 shows effects of…

Line 225          table 4 M and F: please one letter in the middle (as in table 3); the same for table  5

Line235                       Table 6 shows

line 294           extensive information? à offers information…

line 295           effect on lipid….peroxidation during storage

line 311           Own data indicated that…

line 335           delete the blank behind chicken

line 351           pH values

line 369           delete the blank behind up the

line 360           The pH-values of the breast meat

line 415           pages?

Line 439          pages? Doi?

Line 458          Nigella sativa

Line 514          dito

Line 515          pages?

Line 516          Nigella sativa

Line 538          dito

Line 539          pages?

Author Response

The authors would like to thank the reviewers for all comments and suggestions which may improve manuscript quality. The authors made changes to the text in line with the reviewers' renewed comments.

Review 2
